# Dynamics of shallow wakes on gravel-bed floodplains: Dataset from field experiments

Oleksandra O. Shumilova[1], Alexander N. Sukhodolov[1], George S. Constantinescu[2], and Bruce J. MacVicar[3]

[1]Department of Ecohydrology, Leibniz-Institute of Freshwater Ecology and Inland Fisheries, Berlin, 12587, Germany
[2]Department of Civil and Environmental Engineering, University of Iowa, Iowa City, IA 52246, USA
[3]Department of Civil and Environmental Engineering, University of Waterloo, Waterloo, N2L3G1, Canada

*Correspondence to*: Oleksandra O. Shumilova (shumilova@igb-berlin.de)

**Abstract.**

Natural dynamics of river floodplains are driven by the interaction of flow and patchy riparian vegetation, which has implications for channel morphology and diversity of riparian habitats. Fundamental mechanisms affecting the dynamics of flow in such systems are still not fully understood due to a lack of experimental data collected in natural environments that are free of scaling effects unavoidable in laboratory studies. Here we present a detailed dataset on hydrodynamics of shallow wake flows that develop behind solid and porous obstructions. The dataset was collected during a field experimental

campaign carried out in a side branch of the gravel-bed Tagliamento River in Northeast Italy. The dataset consists of thirty experimental runs in which we varied the diameter of the surface-mounted obstruction, its solid volume fraction, and the porosity at the leading edge, the object's submergence, and the approach velocity. Each run included: (1) measurements of mean velocity and turbulence in the longitudinal transect through the centreline of the flow with up to 25-30 sampling locations, and from 8 to 10 lateral profiles measured at 14 locations; (2) detailed surveys of the free surface topography; and

(3) flow visualizations and video-recordings of the wakes patterns using a drone. The field scale of the experimental setup, the precise control of the approach velocity, configuration of models, and the natural gravel-bed context for this experiment makes this dataset unique. Besides enabling the examination of scaling effects, these data also allow the verification of numerical models and provide insight into the effects of driftwood accumulations on the dynamics of wakes.

Data are made available as open access via the Zenodo portal (Shumilova et al. 2020) with
DOI https://doi.org/10.5281/zenodo.3968748.

## 1 Introduction

Flow is the governing factor in rivers, influencing redistribution of energy, materials and living organisms within river networks (Sponseller et al., 2013). Natural obstructions present in rivers such as boulder clusters, wood jams, patches of

riparian and aquatic vegetation heterogenize the fluvial environments. Flow patterns evolving around such obstructions are

conventionally referred to as turbulent wakes (Cimbala et al., 1988; Chen and Jirka, 1995; Takemura and Tanaka, 2007; Zong and Nepf, 2011; Chang and Constantinescu, 2015; Chang et al., 2017, 2020). Understanding mechanisms behind their formation will help to address the fundamental questions related to river morphology, sediment dynamic, potential impacts on biota and, consequently, for assessment and restoration of rivers (e.g. Gurnell et al., 2012).

Many environmental flows evolve behind solid obstructions and form confined wakes in which the flow is restricted by the presence of a solid boundary beneath (flow bed) and above (free surface). When the ratio of the flow depth ($h$) to the width of the flow domain ($B$) is approximately less than or equal to 0.2, the flow and the wake are both referred to as shallow (Nezu and Nakagawa, 1993, p. 111). Natural streams are usually shallow and therefore shallow wakes are typical flow patterns in these environments. At large Reynolds numbers (Re = $Uh/\nu$ where $U$ is mean velocity, $h$ is mean flow depth

and $\nu$ is kinematic viscosity) the dynamics of shallow flows are controlled by the bed friction (Chen and Jirka, 1995). Vertical confinement in shallow flows allows for the growth of instabilities generated by topographic forcing or transverse shear into two-dimensional (2-D), large-scale coherent structures with vertical axes of rotation (Rominger and Nepf, 2011). In addition, the interaction of flow with the river bed results in the development of small-scale three-dimensional vortical structures, which affect the large-scale structures and induce a stabilizing effect on the dynamics of shallow wakes. The

effect of bed friction is quantified by a stability parameter $S = c_f D/h$ where $c_f$ is the bed friction factor, and $D$ is diameter of the in-stream obstruction (Ingram and Chu, 1987).

    Natural rivers are characterized by presence of porous obstructions such as woody debris or patches of riparian/aquatic vegetation. Consequently, porosity affects structure and dynamics of wakes formed behind these obstructions. In addition to the occurrence of a von Kármán vortex street typical for shallow wakes behind bluff bodies, these wakes also exhibit a so-

called "bleeding flow" - the flow through the structures (Cimbala et al., 1988; Chen and Jirka, 1995). Relationships between the volume of solid fraction of the obstruction ($\Phi$) and flow pattern formed downstream were studied in the laboratory experiments by Zong and Nepf (2011) and numerical experiments of Chang and Constantinescu (2015). Three flow patterns for assemblages of emerging vertical cylinders were identified: (1) no vortex street associated with the porous obstruction ($\Phi$ < 0.05, Figure 1a), (2) steady wake followed by vortex street ($\Phi$ < 0.15, Figure 1b), and (3) vortex street similar to that

behind a solid body ($\Phi$ > 0.15, Figure 1c). Steady wake (Figure 1b) often can be observed on natural floodplains. This flow pattern is characterized by a region with steady streamwise velocity, which does not change with longitudinal distance behind the obstruction (e.g. Figure 1b shows a steady wake region with delayed formation of a vortex street). This region promotes deposition of particles, which takes a triangular shape as observed in laboratory experiments (Folett and Nepf, 2012 (Figures 5b and 6), Tsujimoto, 1999 (Photo 3)). On river floodplains such depositions of fine sediments more often

take a shape of a half-lemniscate with considerable longitudinal extension that defines the floodplain's morphology (Figure 1d). These depositions also support biodiversity and biogeochemical functioning of floodplain ecosystems (e.g. Bätz et al. 2015; Franzis et al., 2011; Mardhiah et al., 2014).

    Although in recent years there was evident recognition of the importance of wakes for natural fluvial environments (MacVicar et al., 2009; Bertoldi et al., 2011; Gurnell et al., 2012), there is a general lack of detailed field data on such

wakes. Most previous numerical and experimental studies were mainly carried out for idealized geometries with defined characteristics of the obstruction, bed material and channel setup to minimize artefacts related to the scale effect. In contrast, natural obstructions often have inhomogeneous structure (e.g. accumulations of driftwood after floods). Field component was included only in few studies (e.g. Euler et al., 2012; Sukhodolov and Sukhodolova, 2014). With the aim of promoting fundamental hydrodynamic research towards natural fluvial environments, we have launched a joint research program

between four research groups from the IGB (Berlin), MIT (USA), Univ. of Waterloo (Canada) and Univ. of Iowa (USA). This program included conducting field-scale experiments with the following objectives: (1) describing flow characteristics and visualizing wake patterns behind models of solid and porous obstructions; and (2) examining the effects of bed friction on the wake patterns and their dynamics. A related goal is to investigate the effects of small-scale spatial inhomogeneity of the obstruction properties (e.g., as measured by the solid volume fraction) on the wake structure. The design of the field

experiments anticipated that model obstructions will be similar to those used in previous laboratory setups, but up-scaled to correspond to field conditions, thereby allowing direct evaluation of scale effects in the corresponding laboratory datasets.

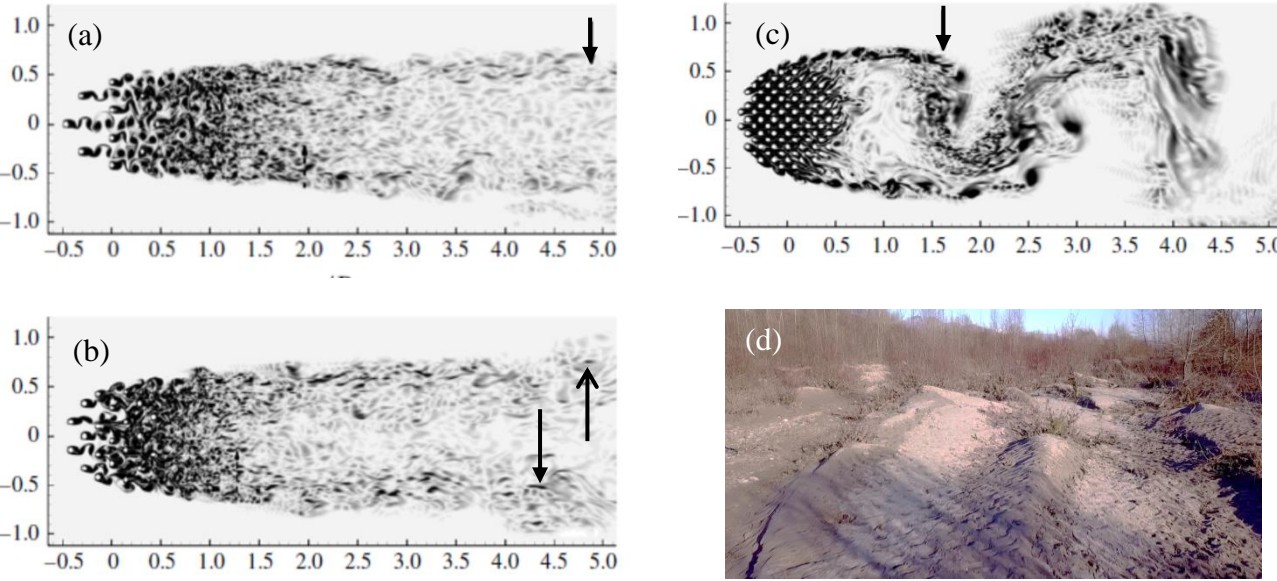

**Figure 1.**  Patterns of vorticity behind a porous obstruction as a function of solid volume fraction (a) $\Phi=0.023$, (b) $\Phi=0.05$, (c) $\Phi=0.2$ (reproduced with permission from Chang and Constantinescu, 2015; the arrows indicate the location of vortex shedding), and (d) the half-lemniscate shaped tale bars behind patches of riparian vegetation on the floodplain of the Tagliamento River.

This paper presents the datasets collected in 2019 on the Tagliamento River in Italy. The data base is accessible via https://doi.org/10.5281/zenodo.3968748 (Shumilova et al., 2020). The data base is composed of raw data converted to conventional ASCII format for key locations, post-processed data and relevant supporting information.

## 2 Field experimental site

Field experiments were carried out on the Tagliamento River in Italy near the village of Flagognia, in the province of Udine. During 2018 the eco-hydraulics group of the Leibniz-Institute of Freshwater Ecology and Inland Fisheries (IGB) started a new research platform named hereafter RIVER-LAB, which succeeds previous initiatives of EAWAG (Switzerland) and the FiRST (Field Research Station Tagliamento) of IGB. Since 2008, IGB has been supporting the research of FiRST by providing the infrastructure, which includes a station living module, storage, laboratory space and equipment and a vehicle for field work and logistics. Besides the research carried out directly by IGB, several international ecological projects have been using the aforementioned infrastructure in the frame of the FiRST. The new RIVER-LAB program primarily aims at conducting eco-hydraulic field-based research, which will aid the development of quantitative theories based on high-quality experimental data obtained in natural environments and thereby free of scaling effects. Data collected in the field is generally characterized by a stronger influence of riverbed topography and roughness, variability of flow structure and flow magnitude, which is not possible to properly reproduce in the laboratory studies.

### 2.1 The Tagliamento River

The catchment of the Tagliamento River is bounded by the Carnian and Julian Alps and stretches through the pre-Alps and Friulian plain southwards towards the Adriatic coast (46°N, 12°20′E). The total area of the catchment is 2580 km$^2$; the length of the main river channel is about 114 km. The hydrologic regime of the river is flashy pluvio-nival with an average total annual runoff about 4.73 km$^3$ at Pinzano and average discharge of ca. 90 m$^3$ s$^{-1}$ at Pioverno (30 km upstream of Pinzano) (Tockner et al., 2003). Because the southern parts of the Alps receive ample rainstorms, the torrents with steep slopes in the Alps are subjected to intensive erosion – a source of coarse grained bedload which is transported by the river during floods and supports the braided channel morphology in the central part of the catchment (Spaliviero, 2003). Although the river is considered to be the last morphologically intact river in the Alps (Müller, 1995; Tockner et al., 2003), it is also subjected to a range of human influences including water obstruction in the upper catchment, organic and plastic pollution, and gravel mining.

### 2.2 In-stream flume

A Tagliamento side branch located at 46°12′9.45″N, 12°58′14.55″E is an ideal site for conducting field-based experiments (Sukhodolov and Sukhodolova, 2014; Sukhodolov et al., 2017; Sukhodolov and Sukhodolova, 2019). During base flow conditions, a gravel bar blocks surface flow from the main channel, which means that the branch is fed only by perennial springs. This makes the hydraulic regime of the field site extremely stable during the summer and mid-autumn seasons. Gravel bed channel has an average depth of $h = 0.35$ m with a bed surface having a slight lateral tilt of 0.5° with respect to the horizontal plane. Bed material has $D_{50} = 22.5$ mm ranging from medium ($D_{16} = 14$ mm) to coarse gravel ($D_{84} = 38.7$ mm) (Sukhodolov and Sukhodolova, 2019).

A 50-m long, 10-m wide rectangular in-stream flume was built in the central part of this side branch. The branch of the stream at the left wall was sealed by an impermeable wall. In the right branch of the stream, a needle weir constructed from plastic plates leaned against a wooden frame was installed (Figure 2a). The approach flow velocity in the in-stream flume was regulated by adding or reducing number of plastic plates, which allowed controlling the degree of flow obstruction.

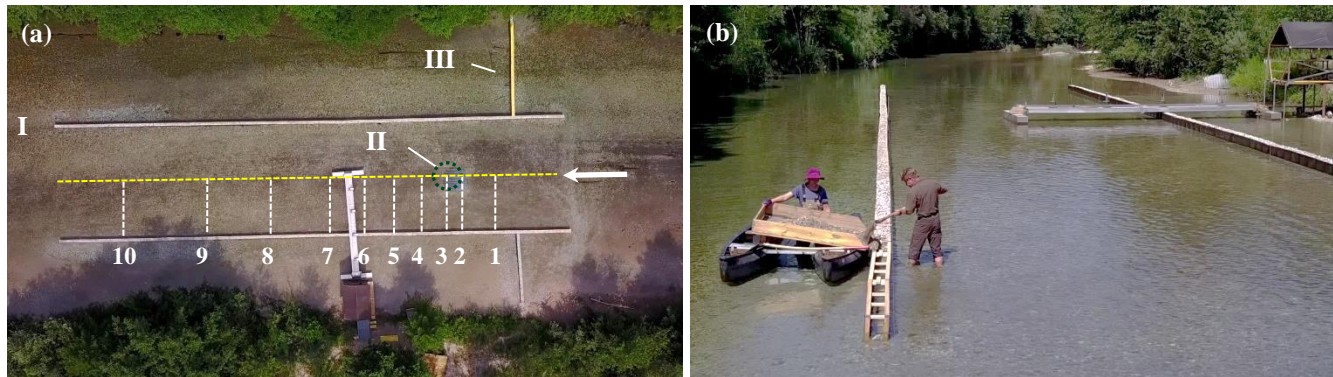

**Figure 2.** In-stream flume (a) aerial view of the flume (I is the wall of the flume, II is a model obstruction, III is a needle weir; the yellow dashed line is the longitudinal transect and white dashed lines are locations of lateral profiles in a typical layout of experimental run), and (b) construction of the in-stream flume

The side walls of the flume were assembled of 1.5×0.5×0.25 m gabions filled with gravel (Sukhodolov, 2019). The frames of the gabions were made of wooden bars and their sides lined with the wire mesh wrapped by plastic covers preventing lateral seepage of fine sediments. Gabions were connected by wooden bars when placed into position during construction (Figure 2b).

**2.3 Models of obstructions**

The obstructions used in the field experiments were porous circular objects with diameters of $D = 1$, 2, and 4 m (Figure 3a), The objects were composed of circular dowels with length ($l$) of 0.40 m and diameter ($d$) of 0.04 m arranged in a staggered formation (Figure 3b). The choice of the diameter of model obstructions was made based on preliminary topographical surveys of riparian vegetation on the floodplain of the Tagliamento River (measurements of vegetation patches diameter with total station, details are not presented in this manuscript and dataset but planned to be included in the analysis of these experiments). The dowels were attached to PVC baseboards using screws. The obstructions were placed in the central part of the in-stream flume 10 m downstream from the inlet section, at a distance equivalent to about 30 times the flow depth. This distance is sufficient to eliminate entrance affects as shown by previous study completed with the same experimental setup (Sukhodolov and Sukhodolova, 2019). The density of the dowels within the obstruction models is defined by the number of dowels per unit bed area, $n$ (cm$^{-2}$), the frontal area per unit volume, $a = nd$ (cm$^{-1}$) and the average solid volume fraction, $\Phi =$

$n\pi d^2/4 \approx ad$ (for circular cylinder elements) (Zong and Nepf, 2011). In our study, we tested porous obstructions with $\Phi = 0.10$. This choice corresponds to cases with equivalent $\Phi$ examined in the most details as part of the laboratory experiments of Zong and Nepf (2011), which thereby allows a direct examination of up-scaling issues. To model the effects of deposition of driftwood at the upstream edge of the obstruction, we conducted several experiments where the solid fraction volume was increased locally by substituting larger diameter dowels ($d = 8$ cm) in the three upstream edge rows of the obstructions. After completing the experiments with emerged obstructions, the dowels were cut to $l = 0.20$ m and the same set of experiments was performed with submerged obstructions. We also carried out several experiments with $\Phi = 1$ (solid obstruction), which allowed a direct comparison with observations of standard wakes past circular objects in deep flows and shallow flows (Chen and Jirka, 1995).

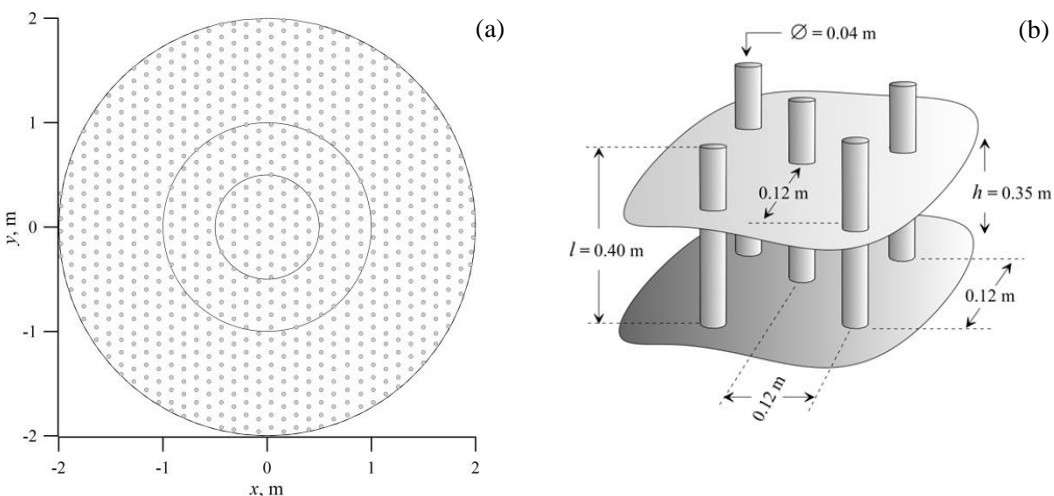

**Figure 3.** Sketch of the obstructions used in the field experiments: (a) locations of dowels (circles indicate boundaries of 1, 2 and 4 m models), (b) size and spacing of dowels (average water depth during experiments was about h = 0.35 m).

## 3 Experimental setup and Instrumentation

All experimental runs were identical in their setup and instrumentation, which were designed for visualizing the flow and measurement of the 3-D flow velocities and the topography of the free surface. The experimental program consisted of systematic variation of the main variables and measurement of the system's responses.

### 3.1 Platform and setup

For measurements of flow velocities and free surface topography a custom-built, floating lateral platform was used. The platform is composed of a 9-m long flat aluminium frame mounted on two custom-built floats made of steel (Figure 4a). During measurements, left side float was placed outside of the flume close to its wall, ensuring that floats will not affect

155  velocity profiles. In addition, both end sides of the platform were stationary fixed to the bottom with the anchoring uprights. To hold the deployment bars of velocimeters or the poles of the topography survey targets during the measurements, seven custom-built deployment mounts were attached to the frame (Figure 4b).

The sensors of the velocimeters were fixed at the ends of 20×20 mm 1.5 m long aluminium deployment bars, which were aligned vertically by the deployment mounts (Figure 4b). Vertical alignment was achieved by 3-axes bubble levels. The

160  orientation of the velocimeter's sensors was aligned such that the y-axis of the sensors was parallel to a cable stretched in a cross-section between locations permanently marked on the railings on the opposing side walls (Figure 4b). The sensors of the velocimeters were located at fixed positions spaced by 70 cm between each other.

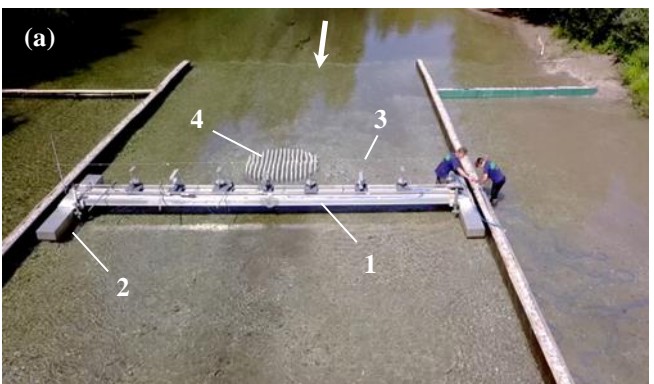 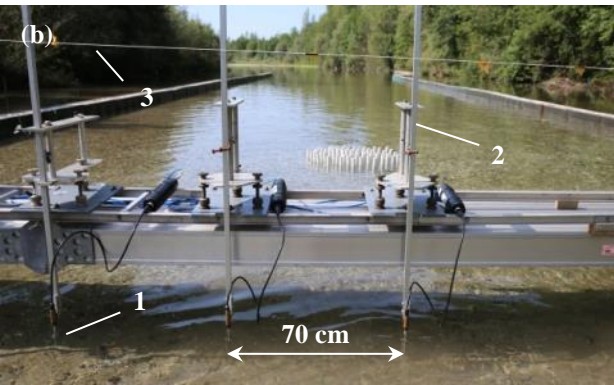

165  **Figure 4.** Measuring frame and setup: (a) floatable platform (1 is the lateral platform, 2 is a float, 3 is a deployment mount, 4 is a porous emerged obstruction), white arrow shows flow direction, (b) instrumental setup (1 is a velocimeter sensor, 2 is a deployment bar, and 3 is an orientation cable).

**3.2 Velocity measurements**

Measurements of 3-D velocities were performed with an array of seven acoustic Doppler Vectrino+ velocimeters with lab

170  probe mounted on 1-m cables manufactured by Nortek AS, Norway (Figure 4b). In each experimental run measurements were performed along a longitudinal transect through the centreline of the circular obstruction ($y = 0$), and at eight or ten lateral transects for the emerging and submerged obstructions, respectively (Fig. 2a). Each longitudinal transect was composed of 25 to 30 sampling locations unevenly spaced from locations situated upstream of the obstruction to locations situated far downstream of the obstruction. Except for the runs with emerged solid obstructions, each lateral transect

175  included 14 sampling locations evenly spaced across the flow at 0.35 m intervals from the centreline towards the left wall of the flume. Within each lateral transect measurements were performed in two sets. In the first set, the first sensor on the platform was aligned with the centreline of the flume. In the second set, platform was shifted 0.35 m to the left side of the flume. Because of the flow symmetry, measurements were performed only on the left side of the setup, similarly to the laboratory experiments carried by Zong and Nepf (2011). At each sampling location, the  probes of the velocimeters were

positioned on a nearly horizontal plane situated 12 cm below the free surface ensuring positioning of the sampling volume in the middle of the water column (distance from probe to the sampling volume is 5 cm). Additionally, in cases with submerged obstructions, velocities were sampled at two vertical locations within the water column (6 cm and 19 cm below the free surface) at locations with strong differences in velocity distribution over depth upstream and immediately downstream of the obstruction. Local flow depth was measured at each sampling location. The instantaneous three dimensional velocity vectors were measured during four to ten minute periods to ensure high quality records of 60 to 500 seconds long (good signal-to-noise ratios, absence of artefacts potentially caused by drifted particles in the proximity of device measurement volume). Because of the low concentrations of seeding particles in the water at the experimental site, fine sediments were dispersed into the stream at a cross-section located 50 m upstream of the entrance of the in-stream flume. This allowed recording velocities with a sampling rate of 25 Hz at most locations. However, at some locations, especially in the zones of high flow instability behind the dowels, the sampling rates were reduced to 10 Hz to reduce possible spikes in the records. These sampling strategies of the measurement program ensured that key characteristics based on velocity measurements were estimated with the accuracy ranging from 3 to 5% of their nominal values (Sukhodolov and Uijttewaal, 2010; MacVicar and Sukhodolov, 2019).

### 3.3 Free surface measurements

The free surface topography was surveyed with an Elta 55 total station (Zeiss, Germany). A reflector target was mounted on the top of a survey pole with a sharp needle tip at its base. The survey pole was mounted in the deployment mounts and levelled to align the vertical position by bubble levels. This ensured its stability and proper orientation during measurements. Before a reading was recorded by the total station, the vertical position of the needle tip was gently adjusted within a roughly 1 minute time interval to account for the slight local fluctuations of the free surface. Replicate measurements were performed when fluctuations were frequent and significant in amplitude (2-3 mm). The total station was located less than 30 m away from the target during the measurements thereby ensuring the accuracy of the free surface measurements was within 1 mm (Sukhodolov et al., 2017). Measurements were performed at 7 locations for each of the lateral transects. The positions of the lateral transects were changed by moving the floatable platform. The density of the lateral transects was high in the vicinity of the obstruction where the gradients of free surface were significant. Finally the surveys of free surface were spatially mapped to obtain the differences in elevations relative to the elevation in the upstream section of the setup. Figure 5 illustrates the free surface topography obtained from the measurements.

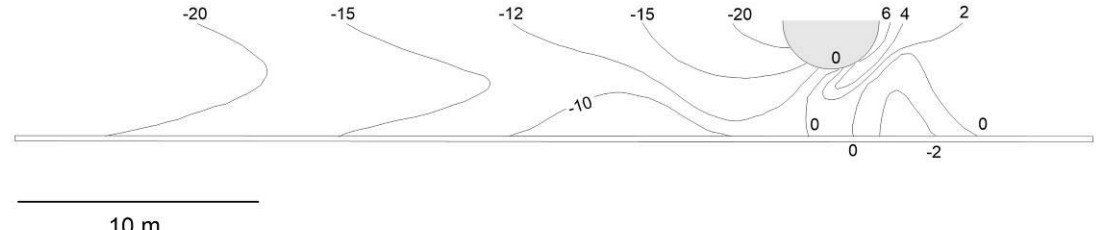

**Figure 5.** Example of the topographical surveys of the free surface in the experimental run 2 (elevations are in mm).

## 3.4 Flow visualizations

For visualization of the flow patterns, two solutions containing uranine (green colour) and rhodamine (red colour) dye were continuously injected through a pair of 5 mm hoses located at both sides of the obstruction. The solutions were supplied to the hose through the constant-head tanks, thereby keeping a constant rate of supply throughout the injection. The solutions were injected for about 15 minutes. Visualizations were recorded by a built-in 4K camera of a Mavic Pro (DJI) quadcopter. In GPS flight-mode with a tripod option, the quadcopter was positioned 40 m above the free surface, near the middle of the in-stream flume. The optical axis of the camera was perpendicular to the water surface (positioned via compass and tilt sensors). The video recording was made with a resolution of 2704x1520 pixels per frame at a rate of 25 frames per second. Examples of flow patterns extracted from video records are shown in Figure 6.

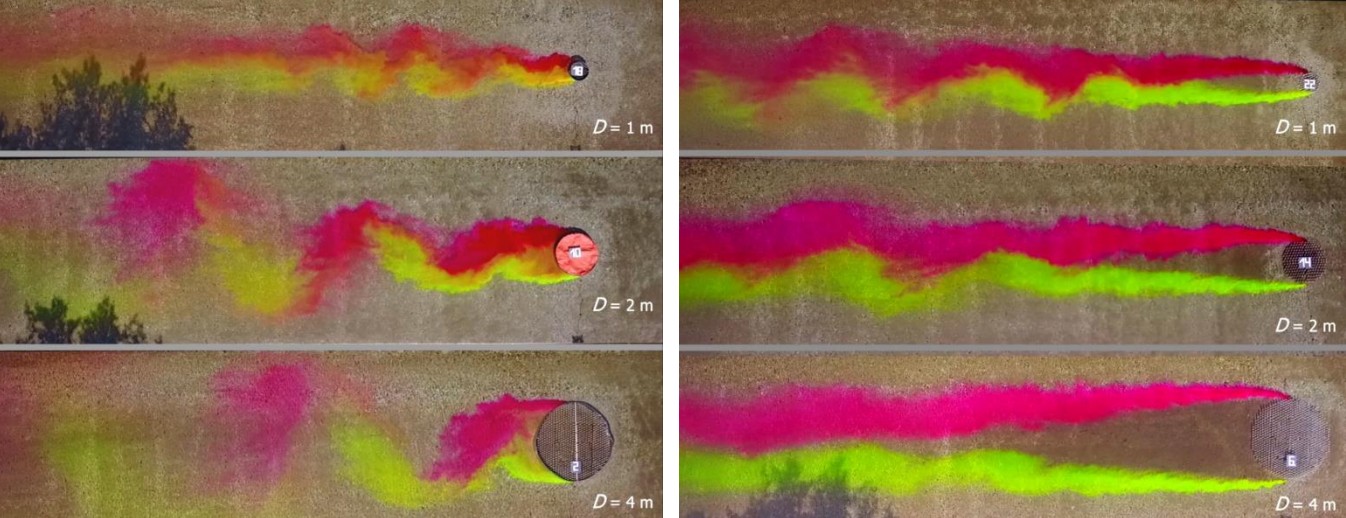

**Figure 6.** Examples of flow visualizations: (a) in runs with an emerged solid obstruction, and (b) in runs with an emerged porous obstruction (figures on the objects show the run number, Table 1).

## 4 Experiments

The experimental program consisted of thirty runs in which approach velocity, diameter of the obstruction, its submergence, and solid fraction volume and the distribution of the solid volume fraction inside the porous obstruction were varied (Table 1). The program was completed within a three-month period from July till October 2019. The hydraulic conditions for the period in which the experiments were conducted were relatively stable and measurements were postponed when fluctuations of the water stage over time exceeded a 2 cm threshold. The average water stage fluctuations for the measured conditions were 1 cm in magnitude.

The experimental runs were identical in terms of instrumentation and measurement protocols, though differed in the amount of collected data (number of longitudinal and lateral sampling locations, for details see section 3.2). Measurement protocols for each run are provided in the respective folders "Supplementary information" in the database.

Each experimental run was complimented by flow visualizations and video recordings of the wake patterns. The topography of the free surface was also measured for each experimental run. The bathymetry was surveyed with high resolution by the total station. Additionally, measurements of mean velocity and turbulence characteristics of the approach flow for both characteristic approach velocities ($U_\infty$) of 0.10 and 0.30 m/s were performed. The lower approach velocity was selected to correspond to the laboratory experiments conducted by Zong and Nepf (2011). The higher velocity was representative of the highest velocity conditions observed at the experimental site. Hydraulic parameters for the approach flow for 0.1 m/s and 0.3 m/s approach were: $Re_{0.1}=0.32\times10^{-5}$ and $Re_{0.3}=0.95\times10^{-5}$, $Fr_{0.1}=0.05$ and $Fr_{0.3}=0.16$, $c_{f0.1}=0.01$ and $c_{f0.3}=0.02$.

**Table 1.** Parameters and variables in the experimental runs

| Run | $D$, m | $\Phi$ | $l$, m | $U_\infty$, m/s | $d$, m |
|---|---|---|---|---|---|
| 1 | 4.0 | 1.0 | 0.4 | 0.1 | 4 |
| 2 | 4.0 | 1.0 | 0.4 | 0.3 | 4 |
| 3 | 4.0 | 1.0 | 0.2 | 0.1 | 4 |
| 4 | 4.0 | 1.0 | 0.2 | 0.3 | 4 |
| 5 | 4.0 | 0.1 | 0.4 | 0.1 | 4 |
| 6 | 4.0 | 0.1 | 0.4 | 0.3 | 4 |
| 7 | 4.0 | 0.1 | 0.2 | 0.1 | 4 |
| 8 | 4.0 | 0.1 | 0.2 | 0.3 | 4 |
| 9 | 2.0 | 1.0 | 0.4 | 0.1 | 4 |
| 10 | 2.0 | 1.0 | 0.2 | 0.3 | 4 |
| 11 | 2.0 | 1.0 | 0.2 | 0.1 | 4 |
| 12 | 2.0 | 1.0 | 0.2 | 0.3 | 4 |

| | | | | | |
|---|---|---|---|---|---|
| 13 | 2.0 | 0.1 | 0.4 | 0.1 | 4 |
| 14 | 2.0 | 0.1 | 0.4 | 0.3 | 4 |
| 15 | 2.0 | 0.1 | 0.2 | 0.1 | 4 |
| 16 | 2.0 | 0.1 | 0.2 | 0.3 | 4 |
| 17 | 1.0 | 1.0 | 0.4 | 0.1 | 4 |
| 18 | 1.0 | 1.0 | 0.4 | 0.3 | 4 |
| 19 | 1.0 | 1.0 | 0.2 | 0.1 | 4 |
| 20 | 1.0 | 1.0 | 0.2 | 0.3 | 4 |
| 21 | 1.0 | 0.1 | 0.4 | 0.1 | 4 |
| 22 | 1.0 | 0.1 | 0.4 | 0.3 | 4 |
| 23 | 1.0 | 0.1 | 0.2 | 0.1 | 4 |
| 24 | 1.0 | 0.1 | 0.2 | 0.3 | 4 |
| 25 | 4.0 | 0.1 | 0.4 | 0.3 | 8 |
| 26 | 4.0 | 0.1 | 0.2 | 0.3 | 8 |
| 27 | 2.0 | 0.1 | 0.4 | 0.3 | 8 |
| 28 | 2.0 | 0.1 | 0.2 | 0.3 | 8 |
| 29 | 1.0 | 0.1 | 0.4 | 0.3 | 8 |
| 30 | 1.0 | 0.1 | 0.2 | 0.3 | 8 |

## 5 Data Processing

Time-series of instantaneous velocities were post-processed with the software package ExploreV (Nortek AS, Norway). Each time-series was visually inspected to identify spikes and discontinuities. Prior to computing basic statistics, 245 discontinuities were removed by cropping the time-series and spikes were replaced with values generated by linear interpolation between adjacent data. Examples of measured time-series in the approach and the wake flows are shown in Figure 7a. The time-series measured in the wake clearly display large fluctuations of the lateral component of velocity with a period of around 20 s (Figure 7a). Calculations of the variance of the velocity components for cumulatively increased sampling durations show that estimates become generally stable for durations exceeding 60 to 120 s for both the approach 250 flow and the wake flow, which shows that the velocity sampling duration of 200 s was sufficient (Figure 7b). The sampling duration and frequency were also sufficient for resolving turbulence spectra at low frequencies (Figure 7c). In the field conditions tested, high levels of background turbulence allowed accurate measurements for the inertial (-5/3) sub-range for sufficiently wide frequency band (Figure 7c).

Post-processing of video records included visual inspection and selection of representative time intervals from 60 to 120 s 255 long. The selection criteria were that the visual pattern was fully developed, the tracer was distributed along the whole flume length, and the overall pattern was representative (Figure 8). The video records were then transformed into a sequence of

JPEG images (1920x1080) with a frame rate of 25 Hz. Ten images separated by 12 s time intervals were then selected for each visualization experiment and stored in the data base for each experimental run. The bench marks on the walls of the models are clearly visible on all images (Figure 8). The locations of the bench marks were accurately surveyed with the total station. By using the bench marks of the pictures in the digitalization procedure, one can assign the exact coordinates from the total station surveys to the locations of the lower right corners of bench marks (Figure 8).

The row surveys of the free surface topography were used to produce contour plots of the free surface. These plots were produced manually because automated contouring software based on kriging interpolation algorithms could not accurately account for the spatial complexity of free surface topography close to the obstruction.

## 6 Data Availability

The dataset described in the study is publicly accessible via https://doi.org/10.5281/zenodo.3968748 (Shumilova et al., 2020).

The data base of velocity measurements is composed of Excel-10 files containing raw data for key locations and the post-processed data for all locations. Supporting materials for each experimental run are also presented in the form of Excel-10 files and JPG/JPEG images.

The post-processed data for each experimental run is saved in a separate file, which name includes the run number. In each file, the data from the longitudinal transects and lateral profiles is contained in a separate worksheet titled with the number of the transect (e.g. LTS is longitudinal transect, and CS-1 is lateral transect N1). The structure of the worksheets is the same for both the longitudinal and lateral transects, and is illustrated in Table 2.

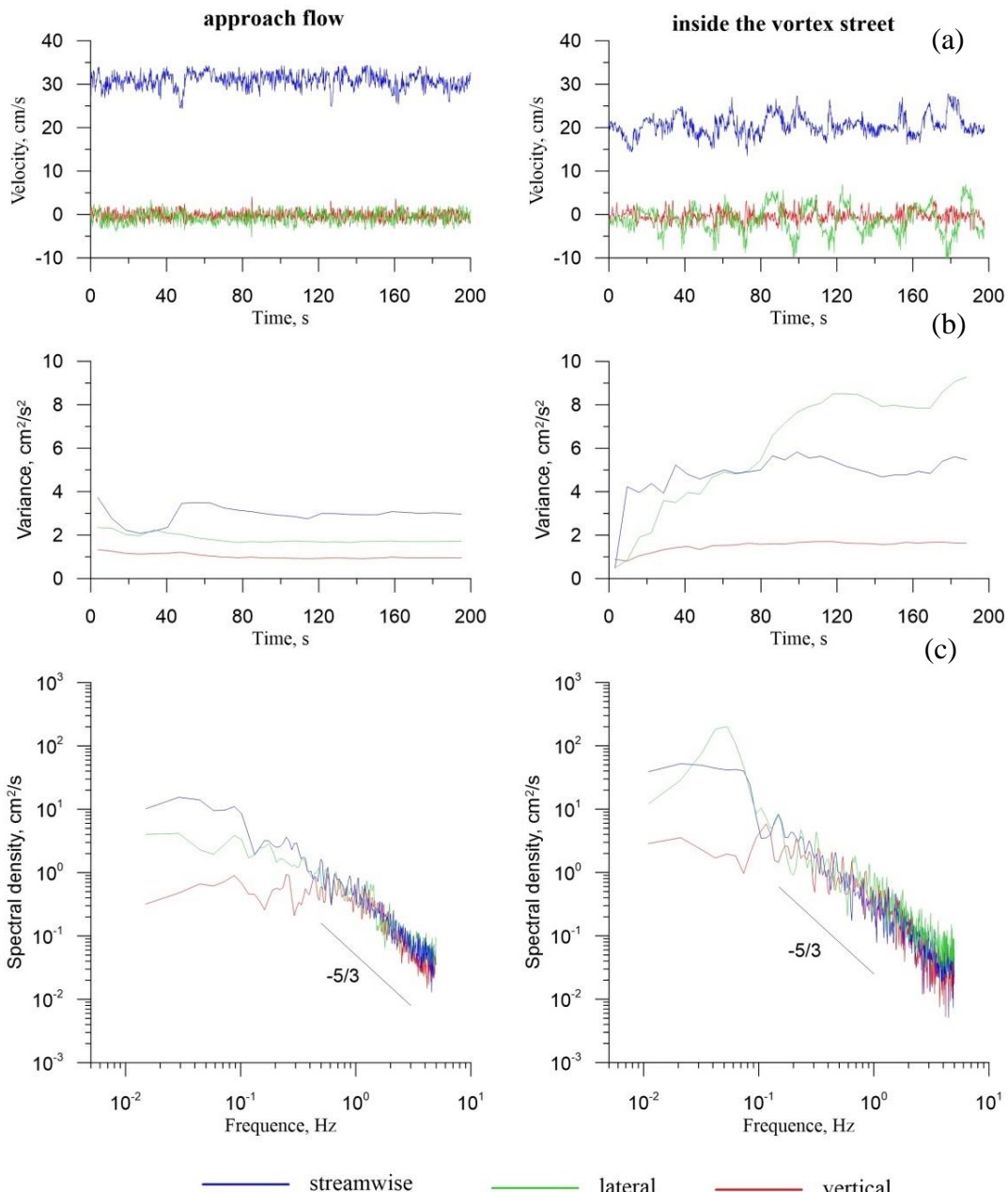

**Figure 7.** Examples of measured velocity data: (a) time series of 3-D velocities, (b) cumulative variances as a function of the band period, and (c) turbulence spectra.

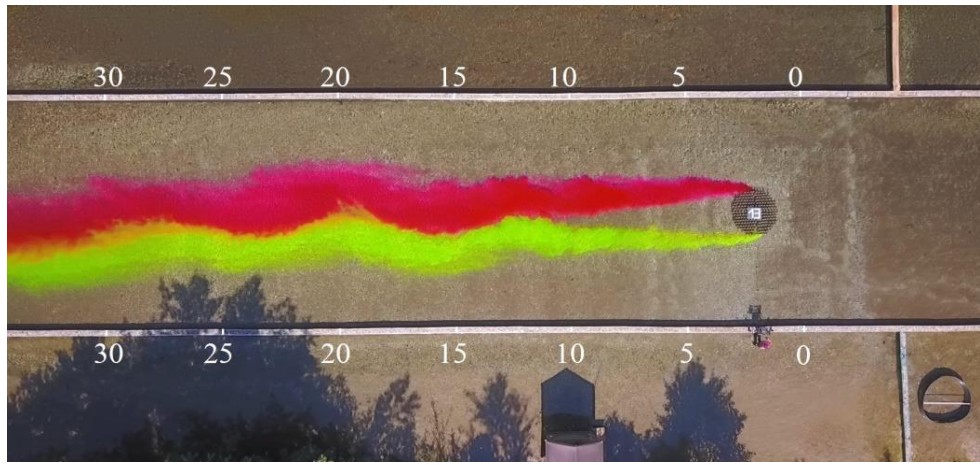

**Figure 8.** Examples of JPEG image recorded in run 13 (numbers on the picture mark the distances in meters for the bench marks on the walls).

Table 2. The structure of post-processed data for velocity measurements in longitudinal and lateral transects.

| nn | $\bar{U}$, cm/s | $\bar{V}$, cm/s | $\bar{W}$, cm/s | $\sigma_u$, cm/s | $\sigma_v$ cm/s | $\sigma_w$ cm/s | $-\overline{u'v'}$, cm$^2$/s$^2$ | $-\overline{v'w'}$, cm$^2$/s$^2$ | $-\overline{v'w'}$, cm$^2$/s$^2$ | $k$, cm$^2$/s$^2$ |
|----|------|------|------|------|------|------|------|------|------|------|
| 1 | 18.8 | -1.3 | 0.2 | 7.2 | 12.4 | 3.4 | 22.0 | 2.8 | -2.3 | 109 |
| 2 | 18.9 | -3.0 | -0.1 | 8.2 | 9.7 | 3.3 | 41.7 | 3.2 | -1.6 | 86.3 |
| 3 | 16.2 | -2.1 | 0.1 | 7.9 | 6.1 | 3.1 | 44.8 | 7.7 | -3.1 | 54.8 |
| .... | .... | .... | ... | ... | ... | ... | ... | ... | ... | ... |
| 14 | 27.7 | 0.5 | 0.9 | 3.1 | 2.0 | 1.3 | -0.4 | 1.5 | -0.2 | 7.7 |

where $\bar{U}, \bar{V}$ and $\bar{W}$ are time-averaged streamwise, lateral, and vertical components of velocity vector, respectively; $\sigma_u, \sigma_v$ and $\sigma_w$ are the turbulent fluctuations of the streamwise, lateral, and vertical components of the velocity vector, respectively ($\sigma_u = \sqrt{\overline{u'^2}}$, $\sigma_v = \sqrt{\overline{v'^2}}$, $\sigma_w = \sqrt{\overline{w'^2}}$); $-\overline{u'v'}$, $-\overline{u'w'}$ and $-\overline{v'w'}$ are the turbulent fluxes of momentum; and $k$ is the turbulent kinetic energy.

The raw data contain time-series of the instantaneous 3D velocities, which are arranged into separate columns containing the time and then the streamwise, lateral and vertical velocities in cm/s. The visualization results for each experiment are presented by sequences of 10 JPEG images sampled at regular moments of time over a 2 minutes period. For each experimental run, the surveys of the free surface are presented as JPG images containing the free surface topography. Hydraulic characteristics of the experiments are presented as Excel-10 worksheets in the supporting materials.

# 7 Conclusions

This paper presents a unique and detailed dataset containing hydrodynamic data, bathymetric and water free surface data, and visualizations of the flow patterns. The dataset is obtained via controlled field experiments completed in a gravel-bed side arm of the Tagliamento River in the Northeast of Italy.

This dataset will help to better understand dynamics of shallow wake flows forming in natural fluvial environments behind patches of riparian vegetation situated on the floodplains. The datasets also provide information on the effect of drift wood accumulation at the leading edges of the patches, which can significantly alter flow dynamics and morphodynamic processes on the floodplains.

    In these experiments, we varied the approach velocity and the diameter, submergence and solid volume fraction at the
leading edge of the model obstruction. Both the scales and roughness of a naturally-formed gravel river bed, together with the characteristic scales of the porous obstruction used in the experiments and the elements forming these obstructions, allow understanding scale effects in data obtained from experiments conducted at laboratory scale. The present datasets also provide comprehensive information on the effects of submergence, which is important given that most vegetation patches on the floodplain become submerged during floods.

The datasets also provide an opportunity for further development and verification of theoretical models by direct and independent assessment of controlling factors. Because of the simplified and idealized configuration of the experimental channel and of the porous obstruction, the present experimental data can be used for verification of advanced numerical models. Another aspect that requires further investigation is the stabilization of large-scale coherent structures by bed friction, which is clearly illustrated by video records of these experiments.

These datasets are presently used in a joint program between the groups of IGB, MIT, IOWA and UW aiming at uncovering the physics of wake forming behind surface-mounted obstructions in a channel from idealized laboratory setups to complex environmental flows at field-scale. In this program the datasets will be primarily used for expanding the theory of shallow wake flows.

*Author contributions.*

All authors took part in development of the study. OS and AS conceptualized the study, developed methods, carried out data collection and data post-processing. OS and AS wrote the original paper with contributions from GC and BM. All authors reviewed and edited the final paper.

*Competing interests.*

The authors declare that they have no conflict of interest.

*Acknowledgements.* This work is a part of the research program "Hydrodynamics of fluvial wakes past in-stream natural objects and their implications for riverbed morphology" granted by the Deutsche Forschungsgemeinschaft (DFG, grant SU 330    405/10).

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
