# Peer review of "Dynamics of shallow wakes on gravel-bed floodplains: Data-set from field experiments"

_Earth System Science Data, 2020_

## Referee Comment (RC1) · Anonymous Referee #1 · 18 Dec 2020

**Review of "Dynamics of shallow wakes on gravel-bed floodplains: Data set from field experiments"**
Shumilova, O.O., Sukhodolov, A.N., Constantinescu, G.S., MacVicar, B.J.

**General Comments**
This paper describes flow velocity measurements, bed and water surface survey data, and flow visualization video of the flow field around circular porous patches recorded in a field flume. Some patch experiments increased the solid volume fraction of the leading edge of the patch in order to approximate driftwood accumulations. The data has been deposited in the Zenodo database.

This paper is well-written and concise, with adequate description of an interesting dataset that will be valuable to future research. While clarification is needed at some points, the manuscript is readable and a useful addition to the field. The flow visualization photos in particular are very well done; the use of simultaneous rhodamine and uranine injections clearly shows the wake interaction even from a drone-held camera and is an improvement over previous methods used for porous patches.

**Specific Comments**
Line 1: This introductory sentence and overall beginning of the paper could be more inviting, especially given the well-written abstract and standard of writing in the rest of the manuscript.

Line 55: Tsujimoto (1999), Follett and Nepf (2012) and Chen et al. (2013, W09517) have all observed downstream fine sediment deposition experimentally. In addition, the diversion of the side streams and eventual downstream join may form a half-lemniscate shape, but fine particle deposition in the steady wake region takes a triangular shape (Tsujimoto 1999, Photo 3; Follett and Nepf 2012, Figure 5b and Figure 6).

Line 53/Figure 1: I cannot identify the vortex street in Figure 1b; visually this seems very similar to Figure 1a.  Please clearly show the delayed onset of the vortex street as described in Line 53.

Line 60: The driftwood aspect (first noticed in Line 118) could be brought out here a bit—otherwise it is difficult to see the difference between "idealized geometries" and the experimental obstructions used.

Figure 2a: what is the wooden apparatus? It is not perfectly perpendicular to the sidewalls and obstruction in the photo; would this have influenced the dataset?

Line 112:  What were the relevant survey results that led you to the choice of obstruction diameter?

Line 114: Do you think this distance was adequate to eliminate entrance effects? Why?

Figure 3: Was the channel shallower near the sidewalls, as it appears in photo? If present, how would the bed variation have affected the velocity profile?

Line 165: how is high quality defined?

Line 192: How did you ensure the camera optical axis was perpendicular to the free surface?

Line 204: Does the 1 cm threshold describe the fluctuations of water stage over time, or space? The variation of the free surface in Figure 5 seems to show 2 cm magnitude variation of the free surface—please clarify.

**Technical Corrections**

Line 35: approximately less than or equal to 0.2

Line 36: therefore (not thereby)

Line 36: prototype does not quite work here--perhaps type, class, regime?

Line 37: consistent use of "the" in the list items

Lines 40-45: Rominger and Nepf (2011) may be appropriate for the sentence ending in …vertical axes of rotation. This paper has a good example of 2D circulations such as those described.

Line 45: The sentence beginning with "Besides…" is confusing. Specifically describing the "features that are typical for shallow wakes behind bluff bodies" would help, e.g. "In addition to the occurrence of a von Kármán vortex street…"

Lines 47-49: This sentence is also vague. Do you mean relatively weaker vortices compared to solid obstructions of the same size? Please be as specific as possible in this section so readers who are not familiar with patch hydrodynamics can understand your later work.

Line 53: The sentence starting with "These wakes are characterized…" is confusing in light of the list in the previous sentence. Is a steady wake present even in 1c? If a vortex street is present "similar to that behind a solid body" (Zong and Nepf 2011 F10b) then why would a steady wake be present? If a steady wake is present in both 1c and 1b, then how do these classifications differ?

Line 92: bed surface

Figure 2a: The obstruction in (II) is not visible easily, perhaps you could circle it or reduce transparency/width of the other marker lines; I assume III is a needle weir (not II as listed); please briefly describe operation of the needle weir; white dashed lines are locations of lateral profiles;

Line 115: isn't $n$ the number of dowels/cm$^2$ and then $a = nd$ =dowels/cm$^2$*cm/dowel?

Line 116: Please clarify that this relationship is true for circular cylinder elements only.

Line 117: "corresponds in the most detail…" please specify that the solid volume fraction is the same "corresponds to cases with equivalent $\phi$ in Zong and Nepf 2011"

Figure 3: The arrow showing flow direction does not look perpendicular to the obstruction. Is this a correct representation of flow direction? If so please discuss the flow angle in the text.

Line 137: Please specify that floats were placed on ends of structure near flume "sides"—I was concerned that float presence may have impacted the flow profile but this isn't the case from F3b

Line 165: four to ten minute periods

Line 207: 'amount of collected data' is vague, please clarify.

Line 209: complemented

---

## Referee Comment (RC2) · Anonymous Referee #2 · 31 Dec 2020

Review of "Dynamics of shallow wakes on gravel-bed floodplains: Data set from field experiments" by Shumilova, O.O., Sukhodolov, A.N., Constantinescu, G.S., and MacVicar, B.J.

General Comments

The paper reports comprehensive field experiments and associated data sets of flow velocities, turbulence parameters, free-surface topography, and flow visualisations involving drone videos. The focus is on the identification of key features of shallow wakes behind the porous obstacles typical for fluvial environments such as floodplains, estuaries, and overland flows on the catchment slopes. The information on such shallow wakes and advanced capabilities for their simulations and predictions is critically important for fluvial hydraulics, morphodynamics, and eco-hydraulics. Practical applications are numerous as the shallow wakes significantly affect hydraulic resistance, mixing and transport of various substances (e.g., sediments, contaminants, and microplastics), habitat fragmentation and most flow quantities important for ecosystem functioning. Thus, the data provided in this paper will help make step changes in current practices related to river and catchment management. Indeed, the coverage of measured parameters, precise experimental set-ups and careful control of background conditions, relevant ranges of experimental scenarios, and the real-life scales of all configurations make these experiments an important steppingstone towards much improved knowledge and predictive capabilities. I must add that as an experimentalist I have been thoroughly impressed by the comprehensive approach, attention to detail, and scientific rigor of this work. It may serve as an excellent example to follow for others in relation to data reporting for a wider use by scientific community.

I found the paper to be very well written and helpfully illustrated with photos, sketches and data plots. The data sets at Zenodo are thoroughly organised, easily accessible, and can be used straight away for various scientific analyses, validation of numerical simulations and modelling, comparisons with the theoretical predictions (both currently available and forthcoming), and, very importantly, for addressing scale issues to help utilizing laboratory experiments for real-life-scale assessments. I believe that the paper and the data sets will be very important contributions to fluvial hydraulics and Earth-Sciences. They can be published, in my view, after a fairly minor revision that should include fixing a few misprints, sharpening text in a few places, adding some minor additional information, and checking/updating a few definitions (please see a complementary file with specific and technical comments and proposed edits).

Specific comments

My specific comments and technical suggestions can be found in a complementary file 'essd-2020-221-supplement.pdf'

Technical corrections

My specific comments and technical suggestions can be found in a complementary file 'essd-2020-221-supplement.pdf'

Please also note the supplement to this comment:
https://essd.copernicus.org/preprints/essd-2020-221/essd-2020-221-RC2-supplement.pdf

[Figure]

**Supplement:**

[revised manuscript text omitted]

---

## Author Comment (AC1) · 6 Feb 2021

**Specific Comments**

Line 1: This introductory sentence and overall beginning of the paper could be more inviting, especially given the well-written abstract and standard of writing in the rest of the manuscript.

Response: We are thankful for the comment. We have updated the first paragraph by adding information on the role of flow for abiotic and biotic processes, as well as on the importance of studying turbulent wakes in shallow flows. Respective references were also added to the reference list.

Original:

Modern science perceives the diversity in fluvial ecosystems in a context of a tight link between spatial heterogeneity in environmental factors and biota. Central to this topic is the flow heterogeneity produced by natural in-stream obstructions such as boulder clusters, log jams, patches of riparian and aquatic vegetation. Flow patterns evolving around these obstructions are conventionally referred to as turbulent wakes (Cimbala et al., 1988; Chen and Jirka, 1995; Takemura and Tanaka, 2007; Zong and Nepf, 2011; Chang and Constantinescu, 2015; Chang et al., 2017, 2020).

Modified:

Flow is the governing factor in rivers, influencing redistribution of energy, materials and living organisms within river networks (Sponseller et al. 2013). Natural obstructions present in rivers such as boulder clusters, wood jams, patches of riparian and aquatic vegetation heterogenize the fluvial environments. Flow patterns evolving around such obstructions are conventionally referred to as turbulent wakes (Cimbala et al., 1988; Chen and Jirka, 1995; Takemura and Tanaka, 2007; Zong and Nepf, 2011; Chang and Constantinescu, 2015; Chang et al., 2017, 2020). Understanding mechanisms behind their formation will help to address the fundamental questions related to river morphology, sediment dynamic, potential impacts on biota and, consequently, for assessment and restoration of rivers (e.g. Gurnell et al., 2012).

Line 55: Tsujimoto (1999), Follett and Nepf (2012) and Chen et al. (2013, W09517) have all observed downstream fine sediment deposition experimentally. In addition, the diversion of the side streams and eventual downstream join may form a half-lemniscate shape, but fine particle deposition in the steady wake region takes a triangular shape (Tsujimoto 1999, Photo 3; Follett and Nepf 2012, Figure 5b and Figure 6).

Response: We added references to the studies that observed particle deposition in the steady wake region in the laboratory conditions. As for the sediment deposition observed by authors on river floodplains, it is difficult to define their form as classical triangular due to higher degree of inhomogeneity in the shape of in-stream objects. Therefore, we specified that on floodplains depositions more often attain longitudinally extended half-lemniscate shape.

Original:

These wakes are characterized by a steady region within which the velocity deficit at the centerline does not change with longitudinal distance. This region promotes deposition of fine

sediments which takes a shape of a half-lemniscate with considerable longitudinal spatial scales that define the floodplain's morphology (Figure 1d).

Modified:

This region promotes deposition of particles, which takes a triangular shape as observed in laboratory experiments (Folett and Nepf, 2012 (Figures 5b and 6), Tsujimoto, 1999 (Photo 3)). On river floodplains such depositions of fine sediments more often take a shape of a half-lemniscate with considerable longitudinal extension that defines the floodplain's morphology (Figure 1d).

Line 53/Figure 1: I cannot identify the vortex street in Figure 1b; visually this seems very similar to Figure 1a. Please clearly show the delayed onset of the vortex street as described in Line 53.

Response: We have showed the onset of the vortex street in Figure 1b with arrows. We also clarified in the main text that for this case the onset is delayed.

Line 60: The driftwood aspect (first noticed in Line 118) could be brought out here a bit— otherwise it is difficult to see the difference between "idealized geometries" and the experimental obstructions used.

Response: we explained what is meant with "idealized geometries" here. We also emphasized that natural obstructions such as accumulation of driftwood on river floodplains have inhomogeneous structure, which is difficult to reproduce during experiments.

Original:

Although in recent years there was evident recognition of the importance of wakes for natural fluvial environments (MacVicar et al. 2009, Bertoldi et al. 2011, Gurnell et al. 2012), there is a general lack of detailed field data on such wakes because most previous numerical and experimental studies were mainly carried out for idealized geometries. Only few such studies included a field component (e.g. Euler et al. 2012).

Modified:

Although in recent years there was evident recognition of the importance of wakes for natural fluvial environments (MacVicar et al. 2009, Bertoldi et al. 2011, Gurnell et al. 2012), there is a general lack of detailed field data on such wakes. Most previous numerical and experimental studies were mainly carried out for idealized geometries with defined characteristics of the obstruction, bed material and channel setup to minimize artefacts related to the scale effects. In contrast, natural obstructions often have inhomogeneous structure (e.g. accumulations of driftwood after floods). Field component was included only in few studies (e.g. Euler et al. 2012, Sukhodolov and Sukhodolova, 2014).

Figure 2a: what is the wooden apparatus? It is not perfectly perpendicular to the sidewalls and obstruction in the photo; would this have influenced the dataset?

Response: it is true that in this photo the lateral structure looks from above non-perpendicular to the sidewalls (we suppose this is what was the questions of a reviewer). This was made on purpose for this photo in order to avoid visual overlap with measurement cross-sections 1-10. During actual measurements the platform was positioned perfectly perpendicular to the side walls and parallel to the predefined cross-sections.

Line 112: What were the relevant survey results that led you to the choice of obstruction diameter?

Response: sentence clarified as follows:

The choice of the diameter of model obstructions was made based on preliminary topographical surveys of riparian vegetation on the floodplain of the Tagliamento River (measurements of vegetation patches diameter with total station, details are not presented in this manuscript and data set but planned to be included in the analysis of these experiments).

Line 114: Do you think this distance was adequate to eliminate entrance effects? Why?

Response: Yes, we think that the distance is adequate to eliminate the entrance effects. The preliminary studies completed with mixing layers with the similar experimental setup on the same river reach (Sukhodolov and Sukhodolova, 2019) show that the riverbed roughness suppresses the lateral growth of a mixing layer at distances around 20h, or about 6 m for this setup. That means that the mixing layers forming at the entrance of the in-stream flume do not grow, but decay at the distances of about 30-40h due to the friction of the riverbed. On the other hand the distance of about 30h is sufficiently long for the bottom boundary layer to be fully developed. Furthermore, the flow entering the flume is already naturally fully developed. We have also mentioned that entrance effects were eliminated in the main text.

Figure 3: Was the channel shallower near the sidewalls, as it appears in photo? If present, how would the bed variation have affected the velocity profile?

Response: The differences in depth across the total span of the in-stream flume were about 5 cm, which is about 10% of average depth. So we expect a slight decrease of flow velocity near the sidewalls, which is more strongly affected by the friction on the walls rather than by depth difference. However, because the walls of the flume were smooth, the sideward boundary layer is quite limited in the lateral extension, which is manifested by the higher degree of lateral homogeneity of flow in the larger portion of the experimental setup.

Line 165: how is high quality defined?

Response: by high quality we mean that measured records had good signal-to-noise ratios and contained no spikes potentially caused by large particles drifted in the proximity and interfering with measuring volume of the devices. We have clarified this also in the main text.

Modified text:

The instantaneous three dimensional velocity vectors were measured during four to ten minute periods to ensure high quality records of 60 to 500 seconds long (good signal-to-noise

ratios, absence of artefacts potentially caused by drifted particles in the proximity of device measurement volume).

Line 192: How did you ensure the camera optical axis was perpendicular to the free surface?

The drone sets the camera flat to the horizontal plain in the lowermost position by adjusting its vertical orientation with compass and tilt sensors. The compass and tilt were calibrated for the specific area before the flights. The flights were performed mainly when the effects of wind were small and the drone had no compensatory tilt for hovering when holding the position. The position for drone was selected at the central location of the setup and the height was adjusted to fit the whole length of setup in the view. Because barrel effect of the lens in this drone camera is excluded the camera had no oblique view and no rectification of images was needed on the post-processing. The accuracy was checked by using about 80 check points geo-referenced with a total station.

Modified text:

The optical axis of the camera was perpendicular to the water surface (positioned via compass and tilt sensors).

Line 204: Does the 1 cm threshold describe the fluctuations of water stage over time, or space? The variation of the free surface in Figure 5 seems to show 2 cm magnitude variation of the free surface—please clarify.

Response: we have clarified that fluctuations of water stage should not exceed a 2 cm threshold over time.

Modified text:

The hydraulic conditions for the period in which the experiments were conducted were relatively stable and measurements were postponed when fluctuations of the water stage over time exceeded a 2 cm threshold.

**Technical Corrections**

Line 35: approximately less than or equal to 0.2

Response: corrected

Line 36: therefore (not thereby)

Response: corrected

Line 36: prototype does not quite work here-perhaps type, class, regime?

Response: "flow prototype" was replaced with "flow pattern"

Line 37: consistent use of "the" in the list items

Response: corrected, "the" was deleted in the list items

Lines 40-45: Rominger and Nepf (2011) may be appropriate for the sentence ending in …vertical axes of rotation. This paper has a good example of 2D circulations such as those described.

Response: thank you for the suggestion, we added the reference.

Line 45: The sentence beginning with "Besides…" is confusing. Specifically describing the "features that are typical for shallow wakes behind bluff bodies" would help, e.g. "In addition to the occurrence of a von Kármán vortex street…"

Response: we corrected sentence beginning with "Besides…" as follows:

In addition to the occurrence of a von Kármán vortex street typical for shallow wakes behind bluff bodies, these wakes exhibit additionally a so-called "bleeding flow" - the flow through the structures.

Lines 47-49: This sentence is also vague. Do you mean relatively weaker vortices compared to solid obstructions of the same size? Please be as specific as possible in this section so readers who are not familiar with patch hydrodynamics can understand your later work.

Response: we have modified the current paragraph to make it clearer for the broad audience not familiar with specifics of patch hydrodynamic. Particularly, we highlighted that porous structure of natural obstructions in rivers (vegetation and woody debris) affect flow patterns behind them. The updated paragraph is given below after the next comment.

Line 53: The sentence starting with "These wakes are characterized…" is confusing in light of the list in the previous sentence. Is a steady wake present even in 1c? If a vortex street is present "similar to that behind a solid body" (Zong and Nepf 2011 F10b) then why would a steady wake be present? If a steady wake is present in both 1c and 1b, then how do these classifications differ?

Response: We have clarified that steady wake is present in the figure 1b. We have also introduced a definition of a steady wake in the text.

Modified:

Natural rivers are characterized by presence of porous obstructions such as woody debris or patches of riparian/aquatic vegetation. Consequently, porosity affects structure and dynamics of wakes formed behind these obstructions. In addition to the occurrence of a von Kármán vortex street typical for shallow wakes behind bluff bodies, these wakes also exhibit a so-called "bleeding flow" - the flow through the structures (Cimbala et al., 1988; Chen and Jirka, 1995). Relationships between the volume of solid fraction of the obstruction ($\Phi$) and flow pattern formed downstream were studied in the laboratory experiments by Zong and Nepf (2011) and numerical experiments of Chang and Constantinescu (2015). Three flow patterns for assemblages of emerging vertical cylinders were identified: (1) no vortex street associated with the porous obstruction ($\Phi < 0.05$, Figure 1a), (2) steady wake followed by vortex street ($\Phi < 0.15$, Figure 1b), and (3) vortex street similar to that behind a solid body ($\Phi > 0.15$,

Figure 1c). Steady wake (Figure 1b) often can be observed on natural floodplains. This flow pattern is characterized by a region with steady streamwise velocity, which does not change with longitudinal distance behind the obstruction (e.g. Figure 1b shows a steady wake region with delayed formation of a vortex street). This region promotes deposition of particles, which takes a triangular shape as observed in laboratory experiments (Folett and Nepf, 2012 (figures 5b and 6), Tsujimoto, 1999 (photo 3)). On river floodplains such depositions of fine sediments take a shape of a half-lemniscate with considerable longitudinal extension that defines the floodplain's morphology (Figure 1d). These depositions also support biodiversity and biogeochemical functioning of floodplain ecosystems (e.g. Bätz et al. 2015; Franzis et al. 2011; Mardhiah et al. 2014).

Line 92: bed surface

Response: corrected

Figure 2a: The obstruction in (II) is not visible easily, perhaps you could circle it or reduce transparency/width of the other marker lines; I assume III is a needle weir (not II as listed); please briefly describe operation of the needle weir; white dashed lines are locations of lateral profiles;

Response: we have highlighted location of the obstruction with additional dashed circle. We corrected that "III" is a needle weir. Principle of the needle weir operation was described in the text a follows:

Additional text:

 In the right branch of the stream, a needle weir constructed from plastic plates leaned against a wooden frame was installed (Figure 2a). The approach flow velocity in the in-stream flume was regulated by adding or reducing number of plastic plates, which allowed controlling the degree of flow obstruction.

Line 115: isn't $n$ the number of dowels/cm2 and then $a = nd$ =dowels/cm2*cm/dowel?

Response: Here $n = \dfrac{1}{cm^2}$,

$$a = nd = \frac{1}{cm^2} * cm = \frac{1}{cm},$$

$$\Phi = ad = \frac{1}{cm} * cm = 1.$$

Line 116: Please clarify that this relationship is true for circular cylinder elements only.

Response: respective sentence was corrected as follows:

The density of the dowels within the obstruction models is defined by the number of dowels per unit bed area, n ($cm^{-2}$), the frontal area per unit volume, a = nd ($cm^{-1}$) and  the average solid volume fraction, $\Phi = n\pi d^2/4 \approx ad$ (for circular cylinder elements) (Zong and Nepf, 2011).

Line 117: "corresponds in the most detail…" please specify that the solid volume fraction is the same "corresponds to cases with equivalent $\phi$ in Zong and Nepf 2011"

Response: respective sentence was corrected as follows:

This choice corresponds to cases with equivalent $\Phi$ examined in the most details as part of the laboratory experiments of Zong and Nepf (2011),…

Figure 3: The arrow showing flow direction does not look perpendicular to the obstruction. Is this a correct representation of flow direction? If so please discuss the flow angle in the text.

Response: We suppose that the reviewer mean the white arrow in the Figure 4a. Here the white arrow is used to help readers visually understand direction of the flow. It does not represent the actual flow angle with regards to the obstruction.

We added respective note to the description of the Figure 4a:

Measuring frame and setup: (a) floatable platform (1 is the lateral platform, 2 is a float, 3 is a deployment mount, 4 is a porous emerged obstruction), white arrow shows flow direction,…

Line 137: Please specify that floats were placed on ends of structure near flume "sides"—I was concerned that float presence may have impacted the flow profile but this isn't the case from F3b

Response: Locations of floats during measurement were specified in the section 3.1. as follows:

During measurements, left side float was placed outside of the flume close to its wall, ensuring that floats will not affect velocity profiles. During measurements both end sides of the platform were stationary fixed to the bottom with the anchoring uprights.

Line 165: four to ten minute periods

Response: corrected as suggested.

Line 207: 'amount of collected data' is vague, please clarify.

Response: we have specified that with amount of collected data we mean number of longitudinal and lateral sampling locations:

The experimental runs were identical in terms of instrumentation and measurement protocols, though differed in the amount of collected data (number of longitudinal and lateral sampling locations, for details see section 3.2).

Line 209: complemented

Response: corrected

---

## Author Comment (AC2) · 6 Feb 2021

Dear Reviewer,

We are very thankful for the review and positive comments regarding contribution of the manuscript to the field.

Please, find below our responses to comments. Relevant changes were also made in the updated version of the manuscript.

Best regards, Oleksandra Shumilova on behalf of the co-authors team

[Figure]

Please also note the supplement to this comment:
https://essd.copernicus.org/preprints/essd-2020-221/essd-2020-221-AC2-
supplement.pdf

**Supplement:**

**Comments and responses**

Line 1: Consider a slightly edited title:

Dynamics of shallow wakes: Data set from field experiments on the Tagliamento River, Italy

Response: Although the suggested change provides a transparent geographical link, we think it might be confusing and the original title is more precise. It is stated in the text that the experimental field site is located on a nameless side branch of the Tagliamento, which has a spring source and during the experiments is not hydraulically linked to the main flow of the Tagliamento. This is important because the flow in the experimental branch is not experiencing the large-scale fluctuations. Furthermore, because of general simplicity of the flow in the experimental setup, the experiments are referring to general situations of flow on gravel bed floodplains and their results are relevant for such systems rather than only for the Tagliamento River.

Line 11: Change "riverbed" to "channel"
Response: corrected

Line 13: Change "present in laboratory studies" to "unavoidable in laboratory studies"
Response: corrected

Line 21: Change "control of the approaching velocity" to "control of the approach velocity"
Response: corrected

Line 30: Change "Turbulent flows…" to "Flow patterns…"
Response: corrected

Line 35: "…the flow and the wake are both referred to as shallow" - I would add a reference to support this criterion.
Response: The threshold value of 0.2 defining narrow open-channel is discussed by Nezu and Nakagawa (1993, p. 111). This criterion is considering the effect of side-walls, which can affect flow in addition to the friction on the bed. Thereby theoretically this is the upper limit for flow shallowness, though for most of shallow flows this ratio is much smaller. For instance, for the dippiest river of the world, the Congo River (h=100 m, B= 2000 m), this ratio is 4 times smaller (0.05) and is close to that in our experiments (0.035). The geometrical

criterion is quite rough and the effect of bed friction on large-scale turbulent structures should be accounted as discussed in the text of that paragraph. We have added the reference.

Line 36: Change "…thereby a shallow wake is the primary flow prototype" to "therefore shallow wakes are typical patterns"
Response: corrected

Line 37: "…$h$ is the mean flow depth and $v$ is the kinematic viscosity" – I would delete 'the' here
Response: corrected

Line 41: "vortical structures with horizontal axes of rotation" – change to "three-dimensional vortical structures". I edited here as real small scale structures tend to be isotropic and therefore do not have preferential orientation.
Response: corrected as suggested

Line 42: Change "shallow flows" to "shallow wakes"
Response: corrected

Line 44: Change "form in natural rivers" to "in natural rivers are formed"
Response: we have modified the paragraph and this combination of words is not present in the updated version

Line 45: Change "include" to "exhibit an additional feature, i.e.,"
Response: corrected

Lines 49-53: The proposed classification is very appealing. However, the threshold values of solidity separating three regimes in general may depend on the internal geometry of 'voids'. A brief clarifying sentence would be useful here, at least it should be mentioned that the proposed threshold values correspond to the assemblages of emerging vertical cylinders.
Response: we have specified in the text that current threshold values were identified for assemblages of emerging vertical cylinders.

Page 2, line 2 after Figure 1: Change "effects of spatial inhomogeneity" to "effects of small-scale spatial inhomogeneity". I added 'small-scale' as at scales l within the range D>>l>>d the flow is 'homogeneous', in terms of spatially averaged quantities.

Response: corrected

Page 2, line 3 after Figure 1: "fractional porosity" - Consider 'solidity' as 'fractional porosity' may be confusing.

Response: we changed "fractional porosity" to "solid volume fraction". Such definition was also used in the abstract of the manuscript.

Page 2, line 5 after Figure 1: "verification of the up-scale effects" – change to "evaluation of scale".

Response: corrected

Line 67: Change "eco-hydraulic" to "eco-hydraulics"

Response: corrected

Line 76: Change "observe" to "to properly reproduce"

Response: corrected

Line 81: "with an average total annual runoff about 4.73 km$^3$ at Pinzano" - It is somewhat unusual quantity to characterize the flow. Why not to use annual mean flow rate? It would be more meaningful in the context of this work, in my opinion.

Response: Thanks for the suggestion. We are not aware of accessible literature sources in English with such information. However, we added information on average discharge at Pioverno, located about 30 km upstream: *The hydrologic regime of the river is flashy pluvio-nival with an average total annual runoff about 4.73 km3 at Pinzano and average discharge of ca. 90 m$^3$ s$^{-1}$ at Pioverno (30 km upstream of Pinzano) (Tockner et al., 2003).*

Line 88: Change "A side branch" to "Tagliamento branch"

Response: corrected as "Tagliamento side branch"

Line 90: Change "river" to "channel"

Response: corrected

Line 93: Change "bad" to "bar"

Response: corrected as "bed surface"

Line 95: Change "of the side branch" to "of this side branch"

Response: corrected

Figure 2: III is a needle weir

Response: corrected

Line 111: Delete "results of"

Response: corrected

Line 117: Change "case examined in the most detail as part of" "most studied case in"

Response: corrected

Figure 3: Please comment on the potential effects of the floats on the measured flow characteristics. Would it be possible to keep the floats outside the flume, just behind side walls?

Response: we have specified in the text that during measurements left side float was placed outside of the flume ensuring that effect on flow measurements was eliminated. In addition, we have specified that during measurements floating platform was at a fixed position.

Line 137: "custom-built floats made of steel" - Please comment on the potential effects of the floats on the measured flow characteristics. Would it be possible to keep the floats outside the flume, just behind side walls?

Response: potential effect of floats was eliminated, see previous comment.

Figure 4: Superb set up. Very impressive. I wonder if floats could be arranged outside the flume?

Response: thanks for the comment. We have specified that left side float was placed outside of the flume during measurements. Within the current setup it was not possible to place both floats outside of the flume as length of lateral platform in this case would not be sufficient.

Line 150: Consider adding one-two sentences at the end of this subsection outlining sampling errors for key characteristics which are based on the velocity measurements and which are included in the data base.

Response: We have added the following sentence: "…These sampling strategies of the measurement program ensured that key characteristics based on velocity measurements were estimated with the accuracy ranging from 3 to 5% of their nominal values (Sukhodolov and Uijttewaal, 2010; MacVicar and Sukhodolov, 2019). "

These estimates include the contribution of acoustic noise, which is about 2-3% and contribution of long-term fluctuations related to variability of hydraulic regime of the stream. The estimates of the impact of acoustic noise are completed using the method of selective integration of the turbulence spectrum implemented in the ExploreV software.

Lines 157-158: "…each lateral transect included 14 sampling locations evenly spaced across the flow at 0.35 m intervals from the centreline towards the left wall of the flume" - Not clear. In line 144 you specified that the distance between Vectrinos was 70 cm. Also, Fig. 4a shows a different set up. Please clarify.

Response: we have clarified current issue as follows:
*Within each lateral transect measurements were performed in two sets. In the first set, the first sensor on the platform was aligned with the centerline of the flume. In the second set, platform was shifted 0.35 m to the left side of the flume.*

Line 164: Change "three dimensional velocities" to "velocity vectors"
Response: corrected

Line 169: "the sampling rates were reduced to 10 Hz to avoid spikes in the records" - I think, the presence of spikes is 'masked' (or 'smoothed') by the reduced sampling frequency rather than fully avoided. Please check.
Response: We agree that it is not possible to fully avoid spikes. We have corrected the respective sentence as follows:
*However, at some locations, especially in the zones of high flow instability behind the dowels, the sampling rates were reduced to 10 Hz to reduce possible spikes in the records.*

Line 179: Change "around" to "in the vicinity of"

Response: corrected

Line 180: Change "high" to "significant"

Response: corrected

Line 199: Please add information in this section on the bulk Reynolds number, Froude number, and friction factor for the approach flow.

Response: information was added as follows:

Hydraulic parameters for the approach flow for 0.1 m/s and 0.3 m/s approach were: $Re_{0.1}=0.32\times10^{-5}$ and $Re_{0.3}=0.95\times10^{-5}$, $Fr_{0.1}=0.05$ and $Fr_{0.3}=0.16$, $c_{f0.1}=0.01$ and $c_{f0.3}=0.02$.

Line 203: Change "3 months" to „three-month"

Response: corrected

Line 203: Change "Hydraulic" to "The hydraulic"

Response: corrected

Line 206: Change "in their" to "in terms of"

Response: corrected

Line 211: Change "patterns" to "characteristics"

Response: corrected

Line 213: Change "with" to "to"

Response: corrected

Line 225: Change "are" to "become"

Response: corrected

Line 225: Change "of" to "exceeding"

Response: corrected

Line 230: Delete "of"

Response: corrected

Line 231: Change "120 s" to "120 s long"

Response: corrected

Line 260, Table 2: Units for stresses should be squared. Please correct.

Response: corrected

Line 261: "$\overline{u}'$, $\overline{v'}$ and $\overline{\overline{w}'}$" - Please consider changing symbols here as conventionally these three quantities are zero if we follow standard definitions.

Response: thanks for the comment, we have corrected this in the manuscript as follows: $\sigma_u$, $\sigma_v$, $\sigma_w$ , specifying that $\sigma_u = \sqrt{\overline{u'^2}}$ etc.

Line 263: "…along the streamwise, lateral, and vertical directions respectively" - delete as this text is confusing, i.e., it is not clear momentum of what components you are writing here about.

Response: corrected